# Maternal effect in salinity tolerance of *Daphnia*–One species, various patterns?

**Andrzej Mikulski** **\*, Danuta Mazurczak**

Department of Hydrobiology, Institute of Functional Biology and Ecology, Faculty of Biology, University of Warsaw, Warsaw, Poland

\* a.mikulski@uw.edu.pl

## Abstract

We experimentally tested the hypothesis that individuals from a single species but genetically different exposed to the same chemical stress factor are able to realize opposite life history strategies–they can invest more resources in current reproduction and release neonates well-prepared to harmful condition or they can invest in their own safety as well as future reproductions and release neonates of poor quality condition. In order to do this, we used the *Daphnia*-salinity model: we exposed *Daphnia magna* females originating from various ponds to two concentrations of sodium chloride, and then observed the key life histories parameters of their offspring exposed or not exposed to salinity stress. Our results confirmed the hypothesis. In a clone from one pond, *Daphnia* exposed to salinity stress produced neonates which were worse-prepared to the local conditions than those released by non-stressed females. In clones from the two other ponds, *Daphnia* released newborns similarly or better-prepared to cope with the salinity stress, depending on the concentration of salt and the duration of their exposure to salinity. Our results suggest that both longer (two-generational) and stronger (higher salt concentration) impacts of selective factors may be perceived by individuals as information indicating reduced chances of successful reproduction in the future and, thus, they may drive mothers to produce better-prepared descendants.

## Introduction

Among the forces determining the phenotype of individual organisms and, in consequence, their fitness [1], population dynamics [2] and microevolutionary processes [3, 4], the role of maternal effect remains one of the key unsolved questions. Each form of influence of maternal phenotype or genotype on the phenotype of descendants is referred to as the maternal effect [5]. In this context, maternal effect is usually interpreted as an intergenerational phenotypic plasticity. Most studies of phenotypic plasticity are dedicated to investigate its adaptive role, enabling the interpretation of results in a broad evolutionary context. In consequence, adaptive maternal effect is, in fact, the most commonly studied of these. Additionally, researchers studying maternal effect have usually tried to find simple cases where mothers who were exposed to adverse environmental factors would produce offspring better prepared to deal with them [6–12]. Current insight into maternal effect requires analysing the intergenerational phenotypic plasticity

**Data Availability Statement:** We have uploaded the dataset from our experiment to a public repository RepOD - https://doi.org/10.18150/KFMBPG.

**Funding:** AM Grant No NN304138940 Polish Ministry of Science and Higher Education. The

funders had no role in study design, data collection and analysis, decision to publish, or preparation of the manuscript.

**Competing interests:** The authors have declared that no competing interests exist.

within the context of maternal fitness, considering not only the fitness of particular descendants but also the number of descendants in subsequent broods or the whole-life reproductive success of mother [13]. Many offspring traits which were previously treated as maladaptive, may in this context be considered to be examples of strategy-increasing maternal fitness.

Generally, in the studies of maternal effect, two types of scenarios have usually been identified, depending on the degree of provisioning of offspring. In the first one, stressed mothers produce poor quality neonates, shifting the cost of living in adverse conditions to offspring [14] and/or they direct resources to the subsequent reproduction [15, 16]. The second scenario, so-called anticipatory maternal effect [17], i takes place when mothers exposed to an environmental threat produce neonates better-prepared to face such conditions than those released by females living in benign conditions. There is also another, until now overlooked, scenario. Stressed females can produce progenies of similar quality as non-stressed mothers (equally prepared to face stressful conditions), change the trade-off between the number and size of the neonates or compensate costs of living under unfavourable environmental conditions by impairing other life history traits (e.g. by delaying maturity).

The most intriguing question is what drives the maternal selection of a particular strategy. The likelihood of breeding again in the future is crucial for this 'decision'–the existing strategies of semelparous and iteroparous organisms significantly differ in this aspect [18]. The chance of reproducing under better conditions in the future is another frequently-discussed issue [17]. It is interesting if the strategy is determined at the species level or it varies between genotypes within a species.

The ideal organismal model to be used in investigating such a problem would be an iteroparic, clonal animal able to adapt to local environmental conditions. This would allow to observe various maternal effect strategies in multiple clones within one species at controlled strength and duration of stress. The biology of the water flea *Daphnia* fully satisfies these demands. This organism is commonly used as a model in studies on phenotypic plasticity, including its intergenerational mode [19, 20]. The role of maternal effect in the expression of the phenotypic reaction of *Daphnia* to environmental factors has been extensively discussed [5, 21] and described in the context of predator-induced shifts in morphology [7], resting egg production induced by deteriorating environmental conditions [22–24] and enhanced tolerance to toxic cyanobacteria [25] and parasitic diseases [26]. Maternal contributions to predator-induced changes in *Daphnia* life history were also demonstrated by Mikulski and Pijanowska [24, 27].

The ideal environmental threat should be one that is common in nature and it should act as an effective selective factor acting proportionally to its concentration, from a small impairment of life history to a lethal effect. Salt seems to be a good choice for such studies. An excess of salinity leads to dehydration of tissues in aquatic organisms, which disturbs many life functions. This effect, as well as the complex mechanisms responsible for tolerance to salinity is well known in *Daphnia* [28]. LC50 (the median of a lethal concentration) of salinity for this cladoceran is about 5.5 g NaCl L$^{-1}$ [29, 30]. Increased *Daphnia* mortality caused by salinity was observed by many authors [29–32]. Under salinity stress, *Daphnia* growth rate [33] and size at first reproduction decrease [34–36], age at first reproduction increases [35, 37] and number of neonates significantly decreases [34, 35]. *Daphnia* demonstrate local adaptations to salinity [38] and salinity can strongly modify the effects of other adverse biotic [39, 40] and abiotic [41, 42] factors affecting *Daphnia* life history.

The main aim of the study was to test (using the *Daphnia*-salinity model) the hypothesis that individuals from one species but different genotypes exposed to the same chemical stress factor are able to realize opposite life history strategies–they can invest mostly in current reproduction and release neonates well-prepared to harmful condition or they can invest in their own safety as well as future reproductions and release neonates of poor quality.

## Methods

In order to test our hypothesis, we aimed at assaying a broad spectrum of *Daphnia* strategies to deal with excessive salinity, and thus we used three clones of *Daphnia magna* from three habitats representing opposite extremes of crucial environmental gradients.

All studied clones were established by hatching resting eggs isolated from the natural environment. The first clone (C1) originated form Binnesee (north-west Germany), which is a large, brackish lake (area 47 790 000 m$^2$, max. depth 3 m) inhabited by fish and occasionally salted by inflow from the Baltic sea (salinity reached 2.5‰). The second clone (TO) was isolated from a small, freshwater, astatic Topiel pond located in Warsaw (area 4785 m$^2$, max. depth 0.45 m). The third clone (KS) originated from the Książęca pond, a larger, freshwater, astatic concreted pond located in Warsaw (area 808 m$^2$, max. depth 0.65 m).

Prior to the experiments, to eliminate the interclonal phenotypic differences that could be caused by a directional maternal effect, animals from all clones were cultured for three generations under constant conditions, the same as in the experiments–individually in 200 ml glass under constant dim light and at a temperature of 20˚C; they were fed green algae *Scenedesmus obliquus* concentrated at 1 mg $C_{org.} \cdot L^{-1}$ (the medium with food was changed daily). The base of the medium was lake water with a low salt content (conductivity below 400 μS/cm, chlorides bellow 60 mg/L), aerated for several weeks and filtered before use (filter size 0.2 μm). The use of natural salt-containing water in the experiment was intended to compare animals from particular treatments to control animals reared in comfortable salinity conditions. Neonates from the second clutch were used to establish the next generation. Neonates released by females from the second clutch of the third generation were split into three groups of 10 individuals each (Fig 1), and placed individually into 200 ml of the appropriate medium. Individuals from the first group were cultured in the control medium. Individuals from the second were cultured in a medium with an addition of low concentration of sodium chloride (3.5 g $NaCl \cdot L^{-1}$). Those from the third group were cultured in a medium with an addition of low concentration of sodium chloride (4.5 g $NaCl \cdot L^{-1}$). All females were transferred to the control medium shortly before releasing their eggs to the brood chambers. Next, neonates produced by three randomly selected females from each maternal treatment were randomly split into the same groups as the mothers (cultured without salt addition, with 3.5 and 4.5 g $NaCl \cdot L^{-1}$). Consequently, nine groups of 10 individuals were obtained which differed in their combinations of maternal and daughter environments. Additionally, ten neonates from each maternal group were used to determine the initial weight of individuals from this generation ($W_0$). Crucial life history parameters were measured: size at birth to a 1 μm accuracy (using NIS-elements Nikon software), age at first reproduction as the age at the moment of releasing first-clutch neonates from the brood chamber (to a 1 hour accuracy), total dry mass at first reproduction of a single female, including mass of the first-clutch neonates (with 0.1 μg accuracy using Orion Cahn C-35 Ultra-Microbalance, Thermo Electron Corporation, USA), number of first-clutch neonates, dry mass of a single neonate in the first clutch (to a 0.1 μg accuracy), and total growth-rate including reproductive investment (i.e. mass of whole first clutch). Before weighing, *Daphnia* were individually placed into aluminium 'boats' and dried for 24 hours at 60˚C. Growth rate $G_j$ was calculated using the formula:

$$G_j = \frac{\ln W_1 - \ln W_0}{t_1 - t_0}$$

Where: $W_0$ –weight of newborn *Daphnia*

$W_1$ –weight of adult *Daphnia* (with first-clutch neonates—max. one hour after they have been released)

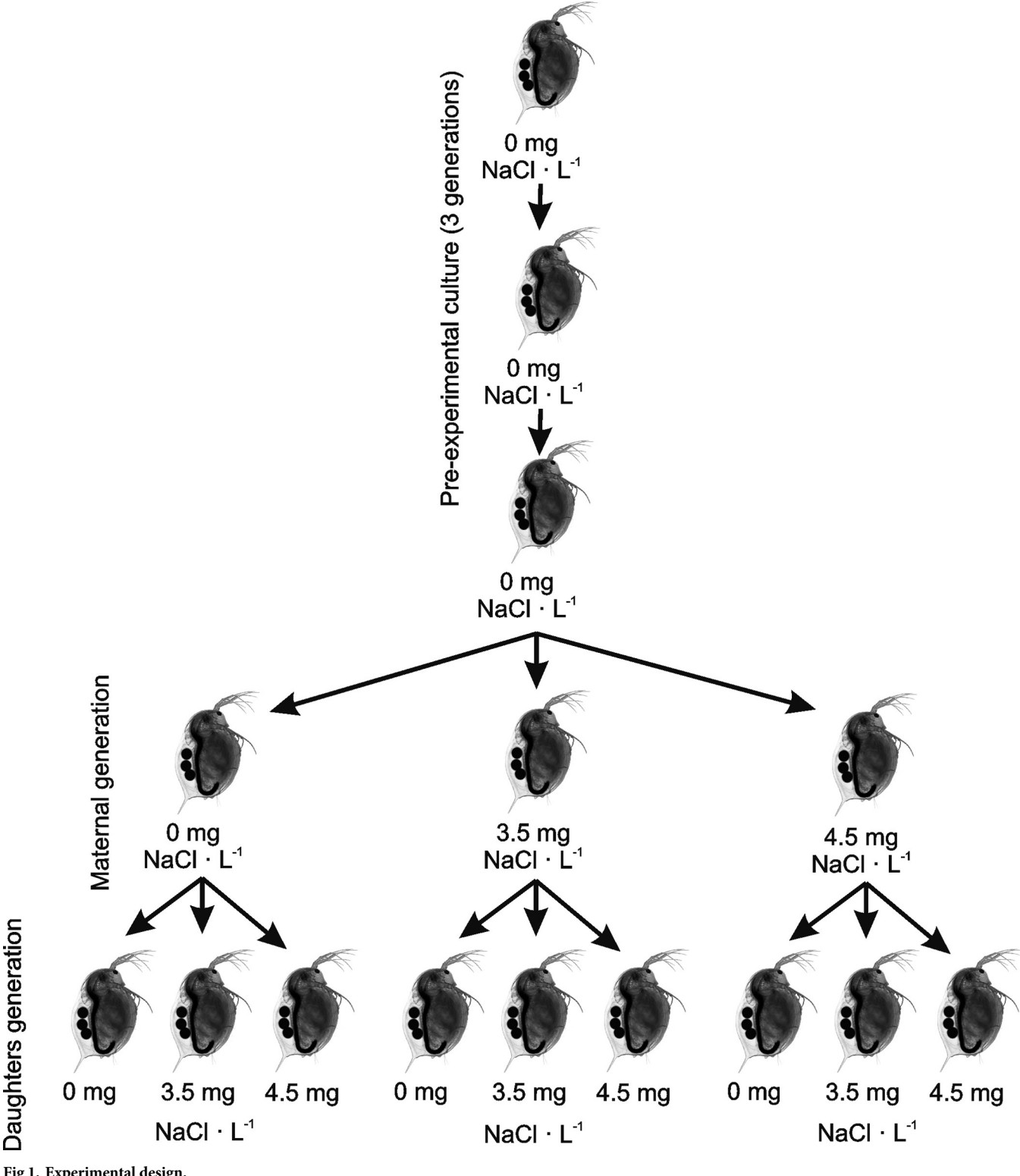

**Fig 1. Experimental design.**

$t_1 - t_0$ time from birth to reach maturity

Juvenile growth rate $G_j$ is a parameter used as a good proximation of fitness in *Daphnia* [43]. Unfortunately, mortality (often associated with toxicity) renders the connection between $G_j$ and fitness less credible (fitness such expressed is overestimated when mortality occurs). In *Daphnia*, $G_j$ may be still better measure of their coping with toxicity than r, because it is *de facto* the measure of net assimilation rate, therefore, it takes into account the direct costs arising from the toxicity. Parameter r is not adequate because it does not consider impairing of assimilation rate under toxicity and costs associated with detoxification. It is very important given that energetic reserves seem to be crucial for fitness under such conditions.

We limited our analysis to the first reproduction of offspring generation because (1) under unpredictable environmental impact, the first reproduction seems to be subjected to the precise optimization (it may be the only opportunity to reproduction), (2) we tried to connect analysis of life history parameters related to reproduction (e.g. age at first reproduction, number of neonates etc.) with analysis of $G_j$. Culturing experimental animals till the second reproduction would make $G_j$ analysis more difficult, and (3) we tried to analyze the same reproduction episode in both generations. The effects of clone, maternal experience of salinity and direct effect of salinity on *Daphnia* life history parameters were tested using a MANOVA model. Next, maternal effect and direct effect of salinity on single *Daphnia* traits were tested using ANOVA with a T-Tukey test (for different N–e.g. Spjotvoll-Stoline test) as a post-hoc.

According to European low, for isolation of resting eggs from public lakes and for invertebrate experiments permits are not required.

## Results

We observed variability in *Daphnia* survivorship during the experiment. Mortality seemed to be stochastic and did not obstruct the analysis of the results. The exception was the group of individuals from TO clone, exposed to the highest salt concentration for two generations, where mortality exceeded 50%. It makes the conclusions less credible in this particular case.

The clonal origin (MANOVA, $\Lambda_{\text{wilks}\ (10,412)} = 0.03$, $P < 0.00001$) as well as direct (via own experience) (MANOVA, $\Lambda_{\text{wilks}\ (10,412)} = 0.26$, $P < 0.00001$) and indirect (via mother's experience) (MANOVA, $\Lambda_{\text{wilks}\ (10,412)} = 0.66$, $P < 0.00001$) effect of the salt concentration significantly affected key parameters of the *Daphnia* life history. We also found a very strong effect for the interactions between all combinations of these factors (MANOVA, $\Lambda_{\text{wilks}\ (40,900)} = 0.57$, $P < 0.00001$).

### C1 clone

Maternal experience of salinity did not affect size at birth of *Daphnia* from clone C1 (ANOVA, $F_{(2,25)} = 1.33$, $P = 0.28173$, Fig 2A), but the direct exposure to salinity had a significant impact on most of the life history parameters.

Both: indirect (ANOVA, $F_{(2,71)} = 2.58$, $P = 0.00003$) and direct (ANOVA, $F_{(2,71)} = 46.32$, $P < 0.00001$) effects of salinity influenced *Daphnia* age at first reproduction in this clone. There was also a strong interaction between these two factors (ANOVA, $F_{(4,71)} = 7.87$, $P = 0.00003$). Experiencing salinity significantly delayed maturity only in animals released by females exposed to a high concentration of salt (Fig 2B).

Indirect (ANOVA, $F_{(2,71)} = 21.49$, $P = 0.00002$) and direct (ANOVA, $F_{(2,71)} = 12.49$, $P < 0.00001$) impacts of salinity affected total mass of *Daphnia* from the C1 clone during first reproduction. There was no interaction between these two factors (ANOVA, $F_{(4,71)} = 2.04$, $P = 0.09785$). Females exposed to high concentrations of salt and released by females exposed to salt were smaller than the others (Fig 2C).

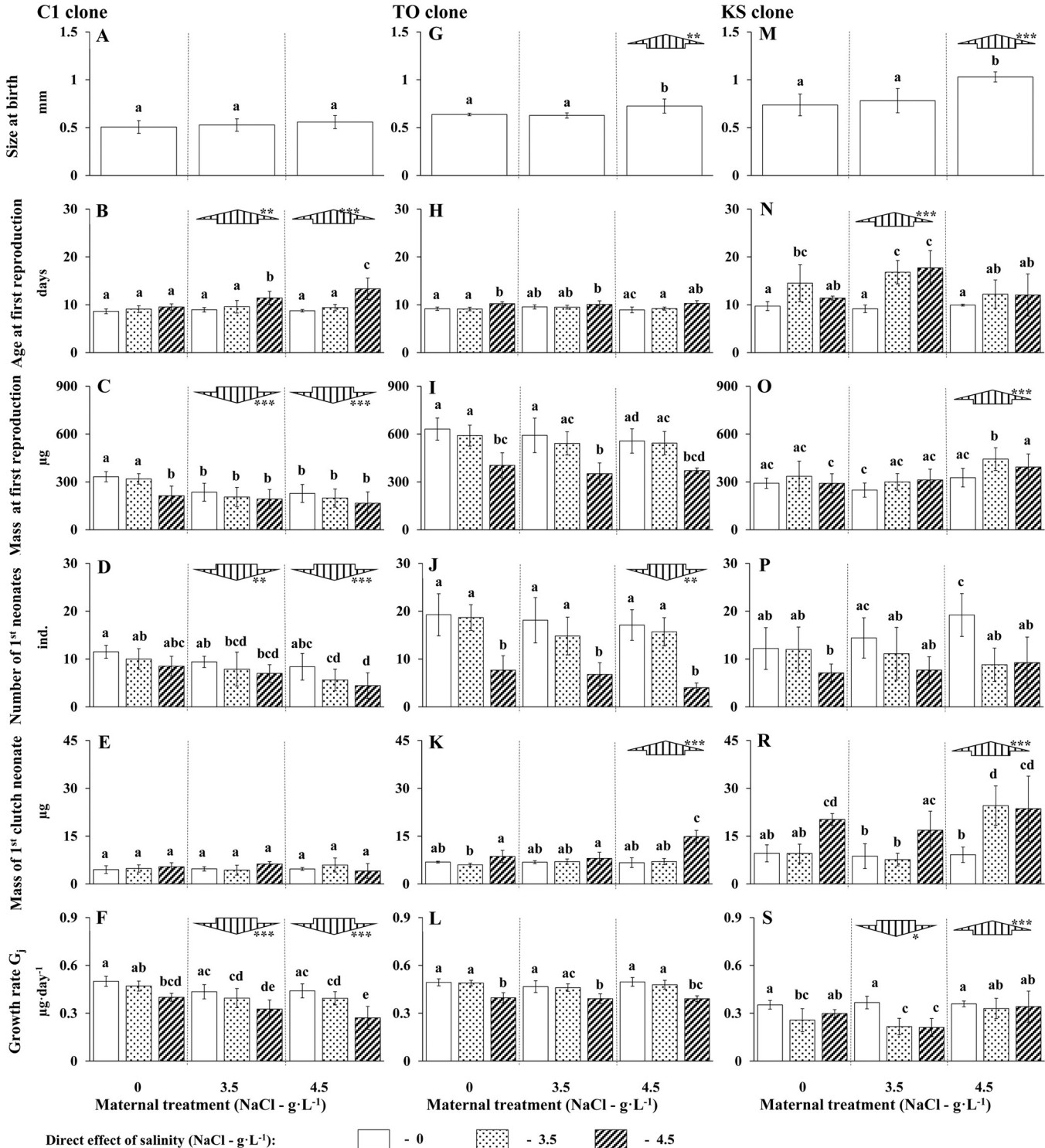

**Fig 2. Direct and indirect (via maternal effect) effects of NaCl on life history parameters of *Daphnia magna* from three clones (averages ± 1 SD); large arrows and their directions indicate significant differences compared to animals born from females not exposed to increased salinity, post-hoc Tukey test calculated only for maternal effect (*—p<0.05, **—p<0.01, ***—p<0.001); small letters indicate homogenous groups, post-hoc Tukey test for different N (Spjotvoll-Stolone test), p<0.05.**

Although maternal (ANOVA, $F_{(2,71)}$ = 13.93, P < 0.00001) and direct (ANOVA, $F_{(2,71)}$ = 12.62, P = 0.00001) experience of salinity affected the number of neonates released by *Daphnia* from the C1 clone, there was no interaction between these factors (ANOVA, $F_{(4,71)}$ = 0.34, P = 0.84936). Both maternal and direct effect of salt similarly decreased the number of newborns released by females (Fig 2D).

None of the factors under study had an influence on the mass of single neonates (maternal effect: ANOVA, $F_{(2,71)}$ = 0.21, P = 0.80937, direct effect of salinity: ANOVA, $F_{(2,71)}$ = 1.28, P = 0.28409). There was, however, a significant interaction between the examined factors (ANOVA, $F_{(4,71)}$ = 4.08, P = 0.00492, Fig 2E).

Maternal (ANOVA, $F_{(2,71)}$ = 26.68, P < 0.00001) and direct (ANOVA, $F_{(2,71)}$ = 48.96, P < 0.00001) effects of salt significantly influenced *Daphnia* growth rate from the C1 clone, but these two factors did not interact (ANOVA, $F_{(4,71)}$ = 1.47, P = 0.21921). Likewise, in the case of the number of first-clutch neonates, both maternal and direct effect of salt similarly decreased the growth rate of females from this clone (Fig 2G).

## TO clone

Maternal experience of salt had a significant impact on size at birth of females from the TO clone (ANOVA, $F_{(2,20)}$ = 11.29, P = 0.00052). *Daphnia* released by females exposed to high salt concentrations were significantly larger at birth than others (Fig 2G). In relation to other life history traits, maternal effect seems to be much less visible in the TO clone than in the C1 clone.

In the TO clone, maternal experience of salinity did not affect *Daphnia* age at first reproduction (ANOVA, $F_{(2,65)}$ = 1.68, P = 0.19485), and did not interact with the direct effect of this factor (ANOVA, $F_{(4,65)}$ = 1.69, P = 0.16288). Direct effect was significant (ANOVA, $F_{(2,65)}$ = 23.50, P < 0.00001) in delaying the reproduction of animals exposed to high concentrations of salt (it was significant only in the case of *Daphnia* born from non-stressed mothers–Fig 2H).

Similarly, maternal experience of salinity did not affect *Daphnia* mass at first reproduction (ANOVA, $F_{(2,65)}$ = 3.06, P = 0.05360) and did not interact with direct effect of this factor (ANOVA, $F_{(4,65)}$ = 0.25, P = 0.90637). However, direct effect was highly significant (ANOVA, $F_{(2,65)}$ = 41.47, P < 0.00001). *Daphnia* exposed to high concentrations of salt were smaller than others (Fig 2I).

There was significant maternal effect (ANOVA, $F_{(2,65)}$ = 3.78, P = 0.02793) and direct effect of salt concentration (ANOVA, $F_{(2,65)}$ = 61.65, P < 0.00001) on number of neonates but there was no interaction between these factors (ANOVA, $F_{(4,65)}$ = 0.71, P = 0.58763). High concentrations of salt caused nearly a twofold reduction in the number of neonates. Maternal experience of high salinity had a similar, albeit weaker, effect (Fig 2J).

There was a significant effect of maternal (ANOVA, $F_{(2,65)}$ = 22.41, P < 0.00001) and direct (ANOVA, $F_{(2,65)}$ = 56.52, P < 0.00001) effect of salinity on the mass of single neonates from the TO clone. There was also a significant interaction between these factors (ANOVA, $F_{(4,65)}$ = 15.87, P < 0.00001, Fig 2K).

The direct effect of salt influenced the growth rate of *Daphnia* from TO clones (ANOVA, $F_{(2,65)}$ = 60.64, P < 0.00001). Neonates released by females exposed to high concentrations of salt had a lower growth rate as compared to neonates from the control and low salt treatments (Fig 2). Although there was a weak effect of maternal experience of salinity on growth rate of neonates (ANOVA, $F_{(2,65)}$ = 3.75, P = 0.02864), this was not reflected in any of the results of the post-hoc test (Fig 2L).

## KS clone

Maternal experience of salinity strongly impacted size at birth of *Daphnia* from the KS clone (ANOVA, $F_{(2,25)}$ = 21.50, P < 0.00001). *Daphnia* released by females exposed to high

concentrations of salt were significantly larger at birth than others (Fig 2M). The maternal effect on other life history traits of *Daphnia* from this clone was more complex and strongly dependent on the concentration of salt to which the mothers were exposed.

Maternal (ANOVA, $F_{(2,74)}$ = 11.89, P = 0.00003) and direct (ANOVA, $F_{(2,74)}$ = 30.42, P < 0.00001) experience of salinity significantly influenced the age at first reproduction of *Daphnia* from the KS clone. There was also a significant interaction between maternal and direct effect (ANOVA, $F_{(4,74)}$ = 6.14, P = 0.00025). Salinity delayed maturation in *Daphnia* exposed to low and high concentrations of salt, but this was significant only among those neonates released by mothers exposed to low salinity stress (Fig 2N).

Maternal (ANOVA, $F_{(2,74)}$ = 19.26, P < 0.00001) and direct effect of salinity (ANOVA, $F_{(2,74)}$ = 9.17, P = 0.00028) significantly influenced the weight of females at first reproduction, but there was no significant interaction between these factors (ANOVA, $F_{(4,74)}$ = 1.63, P = 0.13578). The presence of salt in low concentrations caused the increase of the weight of reproducing females, but it was significant only among *Daphnia* released by females exposed to high concentrations of salt (Fig 2O).

Maternal effect did not influence the number of first-clutch neonates released by females from the KS clone (ANOVA, $F_{(2,74)}$ = 2.03, P = 0.13805), but there was a significant interaction between the maternal and direct effects of salinity (ANOVA, $F_{(4,74)}$ = 3.11, P = 0.02025). The direct effect of salinity was also significant (ANOVA, $F_{(2,74)}$ = 20.80, P < 0.00001). Maternal experience of the presence of salt increased the number of neonates in non-stressed *Daphnia* (Fig 2P).

The weight of a single first-clutch neonate depended on both maternal (ANOVA, $F_{(2,74)}$ = 20.76, P < 0.00001) and direct (ANOVA, $F_{(2,74)}$ = 36.29, P < 0.00001) effects of salinity. There was also significant interaction between these factors (ANOVA, $F_{(4,74)}$ = 9.55, P < 0.00001). Females exposed to high concentrations of salt released larger neonates. Larger neonates were released by both: highly stressed females and those confronted with low salinity, but born by highly stressed mothers (Fig 2R).

Maternal effect (ANOVA, $F_{(2,74)}$ = 14.41, P < 0.00001), direct effect of salinity (ANOVA, $F_{(2,74)}$ = 23.74, P < 0.00001) and their interaction between these factors (ANOVA, $F_{(4,74)}$ = 5.09, P = 0.00112) influenced the total growth rate of *Daphnia* from the KS clone. Increased levels of salinity caused the reduction of the growth rate which was most visible among the *Daphnia* neonates released by females exposed to low concentrations of salt (Fig 2S), and were not visible in female neonates released by highly-stressed mothers (in these cases, there was no effect of salinity).

## Discussion

The observed reaction of *Daphnia* to salinity was consistent with earlier published data. The mortality observed during the experiment is comparable with that described before under similar salt concentrations [29, 31]. Animals exposed to this stress factor mature later [35, 37] and are smaller at first reproduction [34–36]. They also release fewer neonates [34, 35], but the neonates are similar or larger than neonates released by not stressed females. Size of neonates is the only life history feature not impaired by salinity in our experiment (Fig 2E, 2K, 2R). This result is new and shows a peculiar 'reluctance' of *Daphnia* to transferring to offspring the costs of living under adverse conditions

As our results show, *Daphnia* has no universal mechanism of reacting to increased salinity. Each investigated clone reacted in its own, individual way.

In clone C1, from Binnensee (Fig 2, left panel), individuals from the first generation which were exposed to salinity showed no change in age at first reproduction nor in mass of first

neonates, but their other characteristics were impaired: weight at maturity, number of first-clutch neonates and total growth rate decreased, most often in proportion to the strength of the stress factor. The reaction of animals from the next generation was stronger, and the impairment of their life history traits depended on the level of salinity experienced by the mothers. This was clearly visible in age at first reproduction (Fig 2B)–only neonates released by females exposed to salt delayed maturity when confronted with increased salinity themselves. Thus, in this clone, mothers presumably 'transferred' to offspring the costs of living in an uncomfortable environment.

A different scheme was employed by *Daphnia* from the Topiel pond (TO) (Fig 2 –central panel). Low concentrations of salt had no influence on their life history traits. In a similar manner to the C1 clone, high concentrations of salt caused a decrease in weight at the age of first reproduction, as well as a decrease in the number of first-clutch neonates and the growth rate. However, the contrary phenomenon was also observed–females exposed to high levels of salinity released larger neonates than others (Fig 2K). Generally, individuals released by stressed mothers were not impaired by salinity compared to those released by non-stressed animals. Their reaction to salinity was similar, with one exception: though *Daphnia* exposed for two generations to high salinity modified their reproductive strategy to 'more K'–they limited the number of neonates they released, and produced much larger (almost twice as large) newborns. Apart from the last example, it can be concluded that mothers of TO clone exposed to salinity produce offspring of similar 'quality' to those of non-stressed females (they do not transfer the costs of living in a harmful environment to their offspring). Females giving birth to exceptionally large neonates when exposed to high levels of salinity stress for two generations sanctions the hypothesis that *Daphnia* are capable of releasing offspring better prepared to cope with high levels of salinity than those released by non-stressed mothers. High mortality (up to 60%) observed in this clone (among individuals exposed to the highest salt concentration for two generations) undermines this conclusion. However, the range of values of all measured life history parameters among surviving individuals consistently differed from other treatments which, again, validates this conlusion. Reaction to salinity in the clone from Książęca pond (KS) (Fig 2 –right panel) seems to be the most complex. In the first generation exposed to high levels of salinity, the cost of living in a salty environment was visible among *Daphnia* exposed to low concentrations of salt, delaying the age at first reproduction which, in consequence, resulted in decreased total growth rate. *Daphnia* exposed to high salinity tended to decrease the number and increase the size of their first-clutch neonates, so they react as individuals from clone TO exposed to salinity for one generation longer. Maternal effect also depends heavily on salt concentration. *Daphnia* released by females exposed to low concentrations of salt seemed to bear the costs of the maternal environment, in a similar way to the maternal generation, resulting in delayed maturation and thus a decrease in total growth rate. *Daphnia* released by females exposed to high levels of salinity did not bear such costs, although they prepared their offspring to face harmful conditions (individuals exposed to salinity produced larger neonates than the others, even those which were exposed to low concentrations of salt). Most importantly, salinity did not impair their growth rate (Fig 2S), so they seemed to be much better prepared for living in high salinity conditions than neonates released by females exposed to low concentrations of salt or not exposed to salt at all. Three different maternal strategies of *Daphnia* living under conditions of high salinity were observed in our study. In the first strategy, which was shown by *Daphnia* from the C1 clone, offspring bore the costs that their mothers incurred in harmful environments, being worse-prepared to current conditions than neonates released by non-stressed females. The second strategy was seen in the TO clone, and in those females from KS clone which were released by mothers exposed to low salinity. In this case, females did not transfer the costs of living in harmful environments

to the next generation and produced offspring similarly prepared to the current conditions as those released by females living in salt-free conditions. In the third scenario, observed in the case of *Daphnia* from the KS clone exposed to high levels of salinity, mothers prepare offspring to adverse conditions and produce more fit descendants (which reproduce when achieving larger body sizes and then release larger neonates) than those released by non-stressed females. Choice of maternal strategy does not depend solely on genotype. In a single clone, different strategies may be realized depending on the degree of harmful threat encountered in the environment or the number of generations the organisms are exposed to this threat. The change of maternal strategy under two-generational exposure to high levels of salinity (environmental stress) is so far a poorly-described ecological phenomenon. Longer exposure to salinity stress drives a mother *Daphnia* to better prepare offspring—to living in adverse conditions, an effect similar to that shown by animals reacting to the highest concentrations of salt. Longer exposure to salinity stress and the intergenerational transfer of its costs may be a physiological equivalent of short term exposure to higher levels of a stress factor. On the other hand, long-term exposure can carry information about the persistence of environmental stress and may be perceived by individuals as a forecast of reduced chances for future reproduction (see [17]). As a consequence, longer exposure to abiotic stress may promote increased investment in current reproduction.

Increased investment in current reproduction in maternal strategy is also connected with increasing level of stress factors. Experiencing the sub-lethal salt concentration decreases the probability of survival until the next opportunity to reproduce and promotes greater investment in the quality of neonates from the current reproduction. This observation supports earlier identification of factors determining the adoption of particular strategies by mothers [17, 18].

A weak relationship was found between the size / weight and 'quality' of the offspring. Stressed females should release significantly larger (better equipped) newborns to ensure offspring resistance to stress similar to the resistance of offspring born by non-stressed females (the TO clone). Neonates of sizes similar to those neonates released by non-stressed females turned out to be unable to cope efficiently with increased levels of salinity. Size at birth is not always a good predictor of *Daphnia* fitness and maternal effect is not limited to determining the quantity of resources uploaded to eggs, but also in the transfer of information in the form of hormones and other transcription factors (see e.g. [5]) which can reprogram individual development and adjust it to current conditions.

The adaptive maternal effect was usually interpreted as a kind of "immunization" against specific selection factors which the females predict [6–12]. However, it is a mechanism implemented by mothers, which is hard to analyse properly abstracting from maternal fitness. The adaptive maternal strategy may result in no change in the fitness of a single offspring individual or even a decrease in its fitness. This creates a continuum of strategies involving energy allocation and using various mechanisms to influence the ontogenesis of offspring. Our results show that the choice of such a strategy is not necessarily determined genetically, but may itself be an element of phenotypic plasticity. It also shows the great complexity of optimization mechanisms related to reproduction and the weakness of fitness analyses that neglect the fate of the offspring.

## Conclusions

1. Opposite maternal strategies–investing mostly in current reproduction and release neonates well-prepared to harmful condition or investing in own safety and future reproductions but releasing neonates of poor quality–may be realized in individuals exposed to the

same stress factor and belonging to the same species, but genetically different or belonging to one genotype but exposed to different intensity of the stress factor.

2. Both longer (two-generational) and stronger (higher concentration) impacts of selective factors may be perceived by individuals as information indicating reduced chances of successful reproduction in the future and, thus, they may drive maternal strategies to produce first clutch descendants better-prepared to adverse conditions.

## Acknowledgments

We are grateful to Joanna Pijanowska for her help at each stage of work on the manuscript and to Cleve Hicks for improving the language of the manuscript and valuable substantive comments. We also thank the anonymous reviewers for their comments that led to an improved manuscript.

## Author Contributions

**Conceptualization:** Andrzej Mikulski.

**Data curation:** Andrzej Mikulski.

**Formal analysis:** Andrzej Mikulski, Danuta Mazurczak.

**Funding acquisition:** Andrzej Mikulski.

**Investigation:** Andrzej Mikulski, Danuta Mazurczak.

**Methodology:** Andrzej Mikulski.

**Project administration:** Andrzej Mikulski.

**Resources:** Andrzej Mikulski.

**Software:** Andrzej Mikulski.

**Supervision:** Andrzej Mikulski.

**Validation:** Andrzej Mikulski.

**Visualization:** Andrzej Mikulski, Danuta Mazurczak.

**Writing – original draft:** Andrzej Mikulski.

**Writing – review & editing:** Andrzej Mikulski.

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
