## [Decision Letter · Decision Letter 0]

6 Dec 2022

PONE-D-22-25971Maternal effect in salinity tolerance of Daphnia – one species, various patterns?PLOS ONE

Dear Dr. Mikulski,

Thank you for submitting your manuscript to PLOS ONE. After careful consideration, we feel that it has merit but does not fully meet PLOS ONE’s publication criteria as it currently stands. Therefore, we invite you to submit a revised version of the manuscript that addresses the points raised during the review process. Please, consider the issues raised by Reviewer 2 concerning the study focus and its correspondence to the actual approach using the single-clone experiments. Also, address the methodological issues pointed out by both reviewers and broaden the discussion part.

We look forward to receiving your revised manuscript.

Kind regards,

Elena Gorokhova

Academic Editor

PLOS ONE

Journal Requirements:

“AM Grant No NN304138940 Polish Ministry of Science and Higher Education NO”

Reviewers' comments:

Reviewer's Responses to Questions

**Comments to the Author**

1. Is the manuscript technically sound, and do the data support the conclusions?

Reviewer #1: Yes

Reviewer #2: Partly

2. Has the statistical analysis been performed appropriately and rigorously? 

Reviewer #1: Yes

Reviewer #2: Yes

3. Have the authors made all data underlying the findings in their manuscript fully available?

Reviewer #1: Yes

Reviewer #2: No

4. Is the manuscript presented in an intelligible fashion and written in standard English?

Reviewer #1: Yes

Reviewer #2: Yes

5. Review Comments to the Author

Reviewer #1: Dear author,

I read your manuscript with pleasure. It is well designed and very well evaluated and put into the relevant literature context. You describe a very interesting new aspect of maternal effects. However, I have a more conceptual question regarding the cultivation prior to the actual experiment, as this may have had a particular effect on the C1 clone. You describe that they cultured the different clones under constant conditions before applying the salt concentrations. Which culture medium was used here? Was it filtered lake water used or the 0 NaCl medium? And was the salt concentration of the other lakes also checked? Clone C1 is accustomed to salt concentrations (also fluctuating), so the medium without salt could also mean stress. This could also explain the increased number of offspring with 0 NaCl. But this is only a small remark which does not diminish the importance of the paper.

I still have a few smaller comments:

line 79: delete "known" after Daphnia

line 92: "their" instead of "your"

line 96-97: delete sentence

line 98: why did you use resting eggs?

line 106ff: was each individual cultivated single?

line 137: delete Lampert..... the number is enough

Reviewer #2: This manuscript presents an experiment that tests how salinity experienced by mothers influence responses to salinity in their offspring in three clones of Daphnia magna. As such, it contributes to the growing literature on transgenerational phenotypic plasticity. In general the experiment seem to be conducted in an appropriate way (except for some questions I have regarding mass measurements, see below), and my comments mostly relate to the presentation of the study.

First, I was a bit surprised by the way that the study is introduced. It presents the study as a test of a broad hypothesis about differences among populations in how life history strategies (investment in current vs. future reproduction) evolve. However, each of the three populations included in the experiment were represented by only a single clone, preventing conclusions about population differences (responses may well be equally different among clones within a population). Furthermore, no argument is given for why different responses would be predicted among these three populations. Testing a hypothesis requires an ability to reject it, but the present study would not be able to do this (i.e. if an absence of differences among populations had been observed this might well have been because of similar selective pressures in the three populations). As the authors state, the fact that the three clones showed different responses involves a bit of luck (ln. 97). I therefore think the paper would benefit from a rewriting of the introduction to more precisely present the context within which the experiment fits.

The method description was unclear. Particularly this relates to Ln. 112-118. How many individuals per group (ln 112)? Reared individually? Volumes used? How did they know in advance when eggs would be released to brood chambers (Ln. 115)? How many neonates were used from each mother (ln 116)?

For mass of adults and neonates (Ln. 122-125), were these wet masses? If so, how repeatable are such measurements (given that they have excess water on their bodies)? And additionally, the accuracy of the measurement is given as 10 mikrogram, which is on the same order of magnitude as the presented mean weights for neonates. I suspect measurement error is substantial here, particularly for the C1 clone where the mean neonate mass is smaller than the measurement accuracy, and where it is not surprising that the study is unable to find an effect of maternal treatment. Finally, it says that these are weights of “a single female” and a “single neonate”. This is unclear, really difficult to understand what the sample sizes are here. Growth rates are also based on these data, which makes me wonder how reliable these results are as well. In conclusion, I would be much more confident in the results of this study if it had removed these results and made their conclusions based on the more reliable data on length at birth, age at first reproduction and clutch sizes. Alternatively, the authors should discuss these issues and how they might have influenced their results and conclusions.

The discussion was mostly a repetition of their own results. It would be more interesting to see how their results relate to previous studies on salinity effects in daphnia, and potentially transgenerational plasticity in a broader sense.

6. PLOS authors have the option to publish the peer review history of their article (what does this mean?). If published, this will include your full peer review and any attached files.

Reviewer #1: No

Reviewer #2: No

---

## [Author Response · Author response to Decision Letter 0]

3 Feb 2023

Reviewer #1: Dear author,

I read your manuscript with pleasure. It is well designed and very well evaluated and put into the relevant literature context. You describe a very interesting new aspect of maternal effects. However, I have a more conceptual question regarding the cultivation prior to the actual experiment, as this may have had a particular effect on the C1 clone. You describe that they cultured the different clones under constant conditions before applying the salt concentrations. Which culture medium was used here? Was it filtered lake water used or the 0 NaCl medium? And was the salt concentration of the other lakes also checked? Clone C1 is accustomed to salt concentrations (also fluctuating), so the medium without salt could also mean stress. This could also explain the increased number of offspring with 0 NaCl. But this is only a small remark which does not diminish the importance of the paper.

Thank you for this comment. Indeed, we did not express ourselves precisely. We wrote that the given amounts of sodium chloride in experiment are an additional element but this statement was obviously confusing. The base of the medium was lake water with a low salt content (conductivity below 400 µS/cm2). This information with broader explanation of reasons for application of such procedure has been added (lines 110-111).

“The base of the medium was lake water with a low salt content (conductivity below 400 µS/cm, chlorides bellow 60 mg/L), aerated for several weeks and filtered before use (filter size 0.2 µm). The use of natural salt-containing water in the experiment was intended to compare animals from particular treatments to control animals reared in comfortable salinity conditions.” (lines 106-110)

I still have a few smaller comments:

line 79: delete "known" after Daphnia

The word has been removed.

line 92: "their" instead of "your"

The word has been changed.

line 96-97: delete sentence

The sentence has been deleted.

line 98: why did you use resting eggs?

Isolating clones from active forms of Daphnia is difficult because the collected individuals may belong to the same clone. Also, the clones are likely to change over time (ageing?). In the case of clones obtained from resting eggs, we know their laboratory age and rearing conditions. We could standardize these conditions and eliminate their potentially differentiating effect on the compared clones.

line 106ff: was each individual cultivated single?

Yes. This information is given in line 107.

line 137: delete Lampert..... the number is enough

The link has been removed. We are very grateful for detecting our mistakes.

Reviewer #2: This manuscript presents an experiment that tests how salinity experienced by mothers influence responses to salinity in their offspring in three clones of Daphnia magna. As such, it contributes to the growing literature on transgenerational phenotypic plasticity. In general the experiment seem to be conducted in an appropriate way (except for some questions I have regarding mass measurements, see below), and my comments mostly relate to the presentation of the study.

First, I was a bit surprised by the way that the study is introduced. It presents the study as a test of a broad hypothesis about differences among populations in how life history strategies (investment in current vs. future reproduction) evolve. However, each of the three populations included in the experiment were represented by only a single clone, preventing conclusions about population differences (responses may well be equally different among clones within a population). Furthermore, no argument is given for why different responses would be predicted among these three populations. Testing a hypothesis requires an ability to reject it, but the present study would not be able to do this (i.e. if an absence of differences among populations had been observed this might well have been because of similar selective pressures in the three populations). As the authors state, the fact that the three clones showed different responses involves a bit of luck (ln. 97). I therefore think the paper would benefit from a rewriting of the introduction to more precisely present the context within which the experiment fits.

We agree with this statement. Indeed, the text may suggest that we hypothesize that local conditions influence the strategy of Daphnia females. Indeed, to verify such hypothesis, several clones from each location should be examined. Our hypothesis was supposed to be simpler and it is faithfully reflected in the title of the manuscript. It concerns the possibility of the existence of different maternal strategies related to copying with salinity stress within a single species. Selecting clones from environments with different salt regimes was intended only to increase the probability of getting the telling results with a small number of clones. It was important considering the fact that experiment was conducted with large attention (e.g. all important moments of ontogenesis were determined with 1 hour accuracy for each individual). Considering the significant ontogenetic discrepancies between animals exposed to different salt concentrations, the experiment was very absorbing. We were lucky to be able to show the expected differences using three clones only. Perhaps using other clones from the same locations would produce a different result. However, using these three clones enabled us to confirm our initial hypothesis. In order to dispel doubts, we have changed the text as suggested. 

In Introduction, we changed sentence:

“It is interesting if the strategy is determined within the species or it may differ depending on the conditions experienced by local populations.” 

to sentence:

“It is interesting if the strategy is determined at the species level or it varies between genotypes within a species” (lines 59-60)

Sentence:

“The main aim of the study was to test (using the Daphnia-salinity model) the hypothesis that individuals from one species but different local populations exposed to the same chemical stress factor are able to realize opposite life history strategies – they can invest mostly in current reproduction and release neonates well-prepared to harmful condition or they can invest in their own safety as well as future reproductions and release neonates of poor quality.” 

to sentence:

“The main aim of the study was to test (using the Daphnia-salinity model) the hypothesis that individuals from one species but different genotypes exposed to the same chemical stress factor are able to realize opposite life history strategies – they can invest mostly in current reproduction and release neonates well-prepared to harmful condition or they can invest in their own safety as well as future reproductions and release neonates of poor quality.” (lines 85-89). 

In Conclusions we changed the sentence: 

“Opposite maternal strategies – investing mostly in current reproduction and release neonates well-prepared to harmful condition or investing in own safety as well as future reproductions and release neonates of poor quality – may be realized in individuals exposed to the same stress factor and belonging to one clone but different local population or belonging to one genotype but exposed to different intensity of this factor.” 

to sentence:

“Opposite maternal strategies – investing mostly in current reproduction and release neonates well-prepared to harmful condition or investing in own safety and future reproductions but releasing neonates of poor quality – may be realized in individuals exposed to the same stress factor and belonging to the same species, but genetically different or belonging to one genotype but exposed to different intensity of the stress factor.” (lines 385-390)

In Abstract we changed the sentence: 

 “We experimentally tested the hypothesis that individuals from one species but different local populations exposed to the same chemical stress factor are able to realize opposite life history strategies” 

to sentence:

“We experimentally tested the hypothesis that individuals from a single species but genetically different exposed to the same chemical stress factor are able to realize opposite life history strategies” (lines 11-13)

The method description was unclear. Particularly this relates to Ln. 112-118. How many individuals per group (ln 112)? 

10 individuals - the information is now added (see below)

Reared individually? 

YES - the information was added (see below)

Volumes used? 

200 ml - the information was added:

 “Neonates released by females from the second clutch of the third generation were split into three groups, 10 individuals each (Fig. 1) and placed individually into 200 ml of the appropriate medium.” (lines 111-113)

How did they know in advance when eggs would be released to brood chambers (Ln. 115)? 

The release of offspring is preceded by changes in their morphology - in particular, two comma-shaped eyes merge into one growing spot. Additionally, female starts specific “perinatal movements”. 

How many neonates were used from each mother (ln 116)?

10 individuals in the variant, i.e. 30 from a single mother. This information is available in lines 120-121:

“Consequently, nine groups of 10 individuals were obtained which differed in their combinations of maternal and daughter environments.”

For mass of adults and neonates (Ln. 122-125), were these wet masses? If so, how repeatable are such measurements (given that they have excess water on their bodies)? 

It was dry mass. Information about this and about how animals were dried has been added (see below).

And additionally, the accuracy of the measurement is given as 10 mikrogram, which is on the same order of magnitude as the presented mean weights for neonates. I suspect measurement error is substantial here, particularly for the C1 clone where the mean neonate mass is smaller than the measurement accuracy, and where it is not surprising that the study is unable to find an effect of maternal treatment. Finally, it says that these are weights of “a single female” and a “single neonate”. 

We are very grateful for pointing this error. It really does look absurd. However, the accuracy of Orion Cahn C-35 Ultra-Microbalance is 1/10 (0.1) µg, not 10 µg (http://www.triadscientific.com/en/products/potentiometric/1352/orion-cahn-c-35-microbalance-thermo-scientific/250293). We corrected the text: 

“…total dry mass at first reproduction of a single female, including mass of the first-clutch neonates (with 0.1 µg accuracy using Orion Cahn C-35 Ultra-Microbalance, Thermo Electron Corporation, USA), number of first-clutch neonates, dry mass of a single neonate in the first clutch (to a 0.1 µg accuracy), and total growth-rate including reproductive investment (i.e. mass of whole first clutch). Before weighing, Daphnia were individually placed into aluminium ‘boats’ and dried for 24 hours at 60ºC.” (lines 126-131)

This is unclear, really difficult to understand what the sample sizes are here. Growth rates are also based on these data, which makes me wonder how reliable these results are as well. In conclusion, I would be much more confident in the results of this study if it had removed these results and made their conclusions based on the more reliable data on length at birth, age at first reproduction and clutch sizes. Alternatively, the authors should discuss these issues and how they might have influenced their results and conclusions.

All data were replicated 10 times. It was not, indeed, described precisely. We weighted ten neonates released by females from each maternal variant and estimated initial mass of daughters (W0). This is generally accepted procedure. All experimental females were dried and weighted soon after releasing next generation neonates. Also, these neonates were dried and weighted soon after being released. It allowed to obtain W1 mass individually for each female. So, Gj was calculated with as much precision as possible, individually for each female from all 9 experimental groups. Otherwise, the statistical analysis would be impossible. 

We tried to improve the text. Among others, we added:

“Additionally, ten neonates from each maternal group were used to determine the initial weight of individuals from this generation (W0).” (lines 121-123)

The discussion was mostly a repetition of their own results. It would be more interesting to see how their results relate to previous studies on salinity effects in daphnia, and potentially transgenerational plasticity in a broader sense.

We added a new paragraph and extended the discussion to place our results in a more general context (lines 374-383).

---

## [Editor Report · Decision Letter 1]

13 Mar 2023

Maternal effect in salinity tolerance of Daphnia – one species, various patterns?

PONE-D-22-25971R1

Dear Dr. Mikulski,

We’re pleased to inform you that your manuscript has been judged scientifically suitable for publication and will be formally accepted for publication once it meets all outstanding technical requirements.

Kind regards,

Elena Gorokhova

Academic Editor

PLOS ONE
---

## [Editor Report · Acceptance letter]

23 Mar 2023

PONE-D-22-25971R1 

Maternal effect in salinity tolerance of *Daphnia* – one species, various patterns? 

Dear Dr. Mikulski:

I'm pleased to inform you that your manuscript has been deemed suitable for publication in PLOS ONE. Congratulations! Your manuscript is now with our production department. 

Kind regards, 

on behalf of

Professor Elena Gorokhova 

Academic Editor

PLOS ONE